# Development of a Deep Learning-Based Prediction Model for Water Consumption at the Household Level

Jongsung Kim [1], Haneul Lee [1], Myungjin Lee [1], Heechan Han [2], Donghyun Kim [3] and Hung Soo Kim [3,*]

1    Institute of Water Resources System, Inha University, Incheon 22201, Korea; kjjs0308@naver.com (J.K.);
     haneul6802@naver.com (H.L.); lmj3544@naver.com (M.L.)
2    Blackland Research and Extension Center, Texas A&M AgriLife, Temple, TX 76502, USA;
     heechan.han@ag.tamu.edu
3    Department of Civil Engineering, Inha University, Incheon 22201, Korea; yesdktpdi@naver.com
*    Correspondence: sookim@inha.ac.kr; Tel.: +82-10-3441-1038

**Abstract:** The importance of efficient water resource supply has been acknowledged, and it is essential to predict short-term water consumption in the future. Recently, it has become possible to obtain data on water consumption at the household level through smart water meters. The pattern of these data is nonlinear due to various factors related to human activities, such as holidays and weather. However, it is difficult to accurately predict household water consumption with a nonlinear pattern with the autoregressive integrated moving average (ARIMA) model, a traditional time series prediction model. Thus, this study used a deep learning-based long short-term memory (LSTM) approach to develop a water consumption prediction model for each customer. The proposed model considers several variables to learn nonlinear water consumption patterns. We developed an ARIMA model and an LSTM model in the training dataset for customers with four different water-use types (detached houses, apartment, restaurant, and elementary school). The performances of the two models were evaluated using a test dataset that was not used for model learning. The LSTM model outperformed the ARIMA model in all households (correlation coefficient: mean 89% and root mean square error: mean 5.60 m$^3$). Therefore, it is expected that the proposed model can predict customer-specific water consumption at the household level depending on the type of use.

**Keywords:** ARIMA model; household-level; LSTM model; water consumption prediction

## 1. Introduction

Water is one of humanity's most essential resources, and water supply facilities are necessary for ensuring an efficient supply of limited water resources. The water supply penetration rate is over 80% in developed countries and less than 50% in developing countries, depending on the level of development [1]. In Korea, the water supply penetration rate was 99.3% in 2019, with the aging water supply network accounting for more than 32% [2]. This has caused various water problems, such as water quality deterioration and leakage, and further aggravated water stress.

Therefore, the focus used to be on establishing water supply facilities, but in recent years, the importance of efficient water resource supply has been acknowledged. Many studies have been conducted on short-term future water consumption prediction, which is vital for efficient water management [3–6]. For example, Alvisi et al. [7] emphasized that water supply should be operated based on future water demand to supply water efficiently. They predicted future water consumption patterns using an autoregressive (AR) model, which is a linear time series model that predicts water consumption for adequate water supply. Atsalakis et al. [8] stated that a large-scale water supply management system is necessary to predict water demand accurately. Therefore, they developed an adaptive neuro-fuzzy inference system (ANFIS) by applying the neuro-fuzzy concept to a linear time-series model and compared the predictive performance to that of the AR model. Paulo et al. [9]

noted that accurate short-term water demand prediction is essential for water managers to make efficient decisions. They proposed the harmony search optimization algorithm based on the autoregressive integrated moving average (ARIMA) model to improve water demand prediction performance. Salah et al. [10] proposed a method for removing noise from water consumption data and used the noise-removed water consumption data and an AR model to predict monthly water demand. Furthermore, Haiyan et al. [11] predicted annual water demand for adequate water supply in Beijing, experiencing a water shortage as a result of rapid urban growth. The uncertain autoregressive (UAR) model, which can consider uncertainty in the AR model, was evaluated by comparing its water demand prediction performance to that of the traditional AR model. Traditional time series analysis models, such as AR and ARIMA models, have a limitation in that they cannot predict nonlinear patterns as they are based on linear patterns of past data.

With the recent development of computing technology, machine learning and deep learning are often used to predict nonlinear data in the hydrology field [12–14]. The machine learning and deep learning models, such as random forest (RF) and long short-term memory (LSTM), which can reflect the nonlinear time series characteristics of water consumption, are being used for predicting water consumption [15–18]. For example, Bougadis et al. [19] highlighted that it is necessary to predict future water demand and expand water supply infrastructures to build an optimal water distribution facility based on accurate water consumption prediction. They evaluated the artificial neural network (ANN) and ARIMA models for predicting potential water demand and discovered that the ANN-based prediction model outperformed the ARIMA model. Furthermore, Manuel et al. [20] stated that it is crucial to predict the water consumption of consumers to provide an efficient water supply, and they used ANN, RF, and support vector regression (SVR) algorithms to predict the water consumption of cities in southeastern Spain. SVR had the best predictive performance among the three machine learning algorithms. Mohammed et al. [21] stated that governments must plan water distribution to plan for sustainable development. Moreover, they used an ANN model to predict future water demand and used various optimization techniques, such as Levenberg–Marquardt (LM) and genetic algorithms (GA), to develop an optimal model. Li et al. [22] predicted daily water demand in Hefei, China, by applying various data-driven models, such as LSTM, SVR, and RF models. The SVR model had a mean absolute percentage error (MAPE) value of 7.66%, the RF model had a MAPE value of 2.64%, and the LSTM model demonstrated the best performance with a MAPE value of 1.36%. External factors must be considered to improve water consumption prediction as water consumption is influenced by nonlinear patterns such as holidays, weekdays, weather features, and human activities [23–26]. However, most previous studies have limitations in that they only used past water consumption data without considering external factors.

Bakker et al. [27] stated that predicting water consumption with only a single predictor can produce accurate results, but prediction errors can occur significantly depending on weather conditions. They used a multiple regression model that considers various meteorological variables to predict water consumption in urban areas. They discovered that prediction errors can be reduced by up to 11% when multiple meteorological variables are considered. Austin et al. [28] used a multiple regression model that considers demographics and weather conditions to predict water consumption in Seattle, Washington. Adam et al. [29] used ANN and multiple linear regression (MLR) models that consider historical water consumption data, humidity, and daily variables to predict daily water consumption in Torun, Poland. They discovered that the MAPE for the MLR model was 2.56%, while that for the ANN model was 2.28%, indicating that the ANN model performed better. Previous studies have revealed that it is essential to consider external factors such as weather conditions and previous water consumption data for accurate water demand prediction.

Smart technology has recently been introduced in the water supply field as a result of the advancement in information and communication technology and the fourth industrial

revolution, and internet of things (IoT) devices such as smart water meters are being used for efficient water management [30–33]. Since data for individual users are collected, water leaks and errors can be easily assessed, and various services can be provided by analyzing water consumption patterns for each user. In the United Kingdom, Asset Management Plan 6 was launched to solve a serious leak problem, and the smart water meter project was promoted for efficient water management [34]. Maria et al. [35] used household water consumption data measured by a smart water meter to predict future water consumption in the United Kingdom. In addition, the effect of weather conditions on household water consumption was analyzed. Xenochristou et al. [36] predicted water consumption for one to seven days based on household water consumption data collected at 30-min intervals in the southwestern part of the United Kingdom. The characteristics of 600 households were identified and divided into five groups, and a gradient boosting machine (GBM) model was used to predict the water consumption of each group.

The smart water grid (SWG) project has been promoted in Incheon Metropolitan City, South Korea, and smart water meters have been installed in urban areas to manage water to solve the water shortage problem [37]. Choi and Kim [38] used a multiple regression model, a multilayer perceptron model, and an LSTM model to predict future water consumption based on hourly household and commercial water consumption data collected through smart water meters. They discovered that the LSTM model has better predictive performance than the other models. Various studies on smart water meters are being expanded to establish an efficient water supply plan and provide various water-related services. However, in South Korea, there are still insufficient studies on smart water meters to predict consumer-level water consumption.

Therefore, this study aimed to develop a model for predicting consumer-level water consumption using data from a smart water meter installed in Yeongjong Island, Incheon, South Korea. In this study we used an LSTM-based model, to predict water consumption prediction at the household level, that is suitable for time series analysis among deep learning models and can consider external factors. We evaluated its predictive performance by comparing it to the ARIMA model, which is a traditional stochastic model.

## 2. Materials and Methods

### 2.1. Study Area

The Republic of Korea is a peninsula located at 33°–38° North and 124°–131° East, with mountainous terrain covering more than half of the region. Korea's rainfall characteristics are influenced by its monsoon climate, rainfall is concentrated in the summer (June–October), and regional variations are based on the topographical characteristics. The study area is Yeongjong Island, which is the sixth-largest island in Korea (total area: 125.7 km$^2$) and a representative area with a water shortage problem. It is located in Jung-gu, Incheon.

The SWG project was promoted in Yeongjongdo Island in 2017 to improve efficient water management, and smart water meters were installed in about 500 households. Figure 1 shows the study area and the location of the installed smart water meters. According to the water-use type, the smart water meters were separated into residential, commercial, and public categories. The residential category was further divided into detached houses and large apartments. This research used a case study to develop a model for predicting consumer-level water consumption, and one representative household for each water-use type was selected and analyzed. In Figure 1, the red circle represents the entire location of the installed smart water meter, the blue marker represents the selected smart water meter, and the orange marker represents a meteorological station that collects weather information.

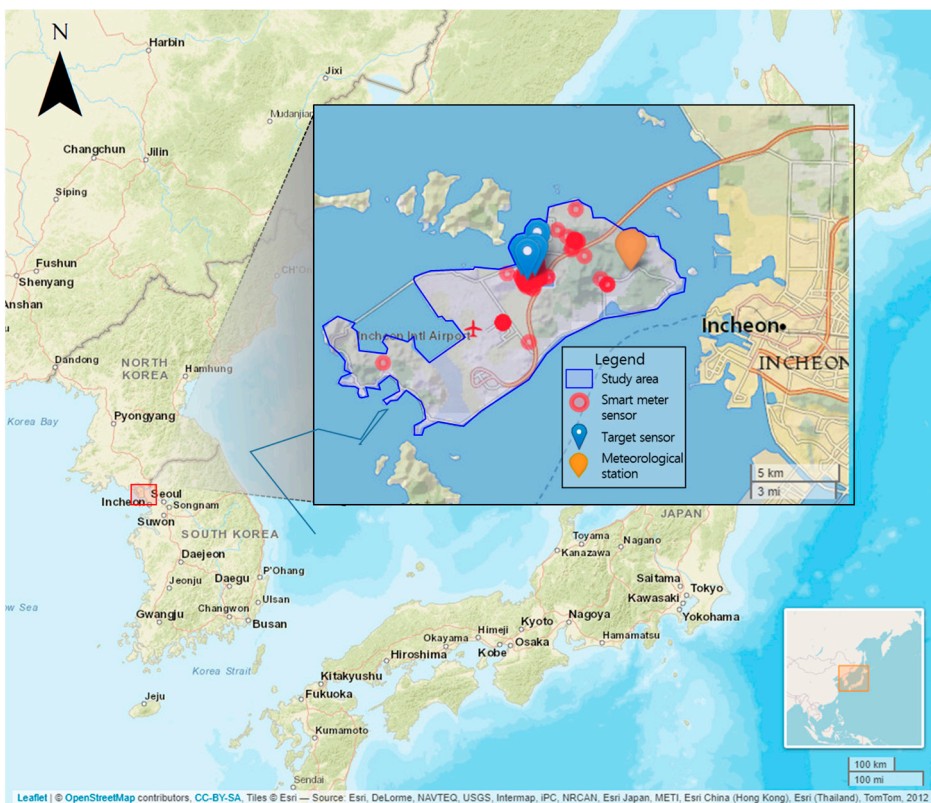

**Figure 1.** The location of the study area and smart water meter.

### 2.2. Data Description

A smart water meter can transmit and receive real-time water consumption and water quality data, which can be used to confirm water consumption fees, leakage, and additional details for each customer.

A smart water meter has been installed in this study area since January 2017 as part of the SWG project, and consumer water consumption data have since been collected. This study used a representative household for each water-use type as the target sensor, and water consumption data were collected from January 2017 to December 2019. Table 1 summarizes the water consumption data for the target sensor based on water-use type. Rows 1 and 2 contain descriptions of the type of use, while rows 3–5 contain water consumption statistics.

**Table 1.** Water consumption data for each water-use type.

| Description | Type A | Type B | Type C | Type D |
|---|---|---|---|---|
| **Use type** | Residential | Residential | Commercial | Public |
| **Detail information** | Detached house (1 household) | Apartment (366 households) | Restaurant | Elementary school |
| **water consumption range (m$^3$)** | 0.5–2.8 | 23.8–214.8 | 0.78–23.7 | 0–55.9 |
| **Mean (m$^3$)** | 1.38 | 153.9 | 10.96 | 24.79 |
| **Standard deviation (m$^3$)** | 0.42 | 30.13 | 2.48 | 9.95 |

Type A represents residential use (detached house), and water consumption is low as the number of users is lower than that for other types. Type B represents residential use (apartment), with the largest number of users and the highest water consumption. Type C represents commercial use (restaurant), and it has the second-lowest water consumption level. Type D represents public use (elementary school) and has the second-highest water consumption level.

Figure 2 shows the weekly water consumption pattern for each water-use type. Types A, B, and C had similar patterns, and their water consumption was high on weekends and low on weekdays. However, the water consumption pattern of Type D was low on weekends and high on weekdays.

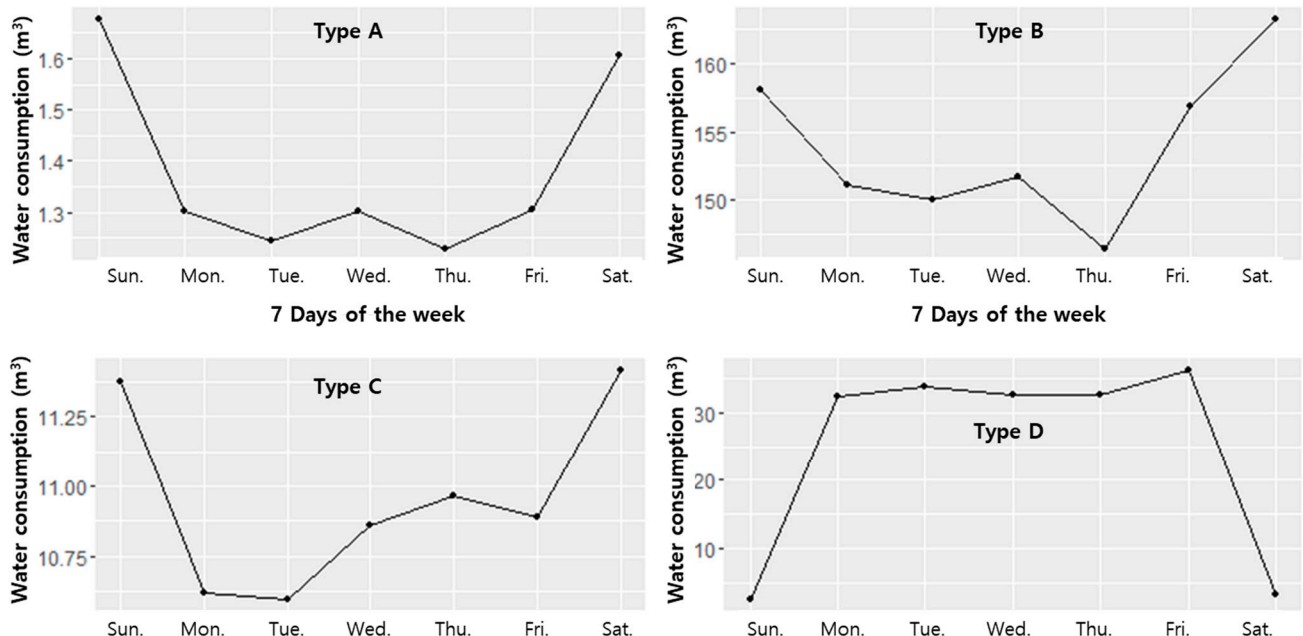

**Figure 2.** Water consumption pattern according to the days of the week.

The reasons for these patterns are as follows: In Types A and B, individuals do not have much time to stay at home as they work on weekdays. In Type C, business is better on weekends than on weekdays depending on the floating population; thus, water consumption is higher on weekends. In Type D, water consumption is high during weekdays as students spend more time at school on weekdays than on weekends. In addition, the difference in water consumption between the days of the week is small in Type C as restaurants are open all days of the week. However, the difference in water consumption between the days of the week is significant in Type D as schools do not open on weekends. In this manner, information on weekdays and weekends is an essential variable for understanding the water consumption pattern by water-use type.

Additionally, weather information was used to identify nonlinear patterns for water consumption based on water-use type. Weather data, such as rainfall, relative humidity, and temperature, were collected at a meteorological station near the study area (orange marker in Figure 1). This station is managed by the Korea Meteorological Administration (KMA), and meteorological data were obtained as daily data from January 2017 to December 2019. Figure 3 shows the monthly time series for each weather condition. Rainfall, relative humidity, and temperature all have maximum values in summer (June to September in Korea) and minimum values in winter (December to February in Korea).

### 2.3. Methodology for Water Consumption Prediction

In this study, the traditional ARIMA model and the LSTM model were used to predict water consumption by consumers, and the performance of the two models was compared. Figure 4 shows the flow chart for this study, including (1) model training and (2) model evaluation. The water consumption data of the selected four types mentioned in Section 2.2 were collected from January 2017 to December 2019. The data were divided into a training dataset (January 2017 to December 2018) for model training and a test dataset (January 2019 to December 2019) for model evaluation. The ARIMA and LSTM models were trained using optimal parameters in the model training phase. The ARIMA model was trained

using past data and the periodicity of the data. However, the LSTM model was trained by considering both the periodicity of the data and external variables such as weather and weekend information. In the model evaluation phase, the prediction results of the two models were evaluated for observed data, and the best-performing model was chosen as the final model. The root mean square error (RMSE) and correlation coefficient (CC) were used as the evaluation metrics in this case.

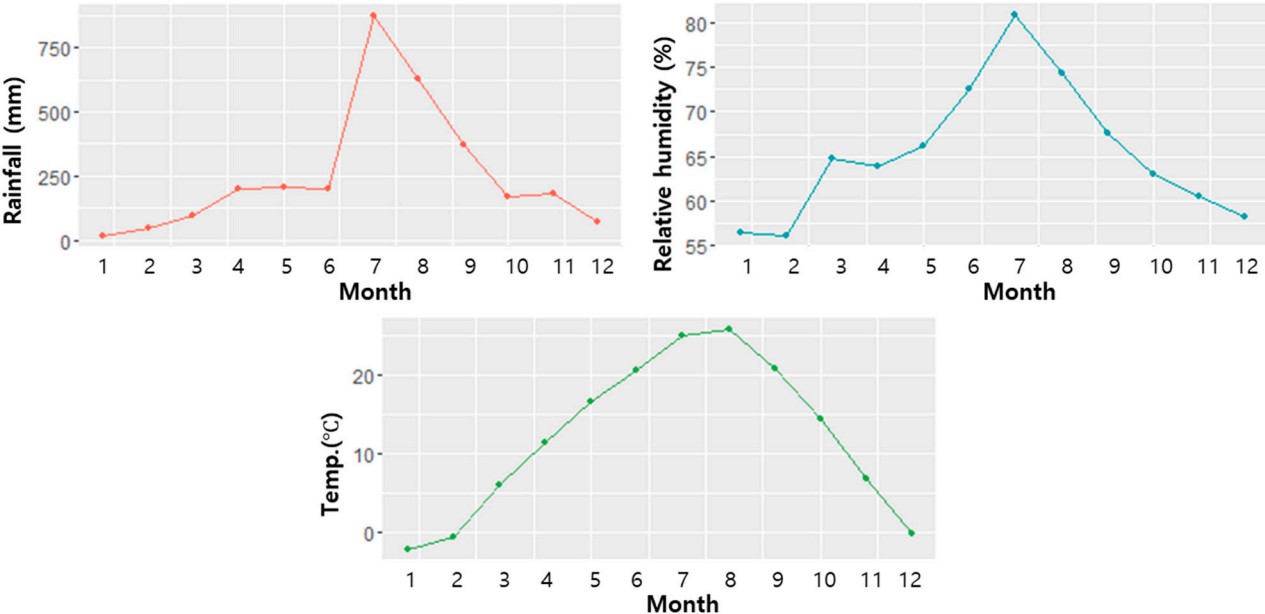

**Figure 3.** Monthly patterns of three meteorological data (rainfall, relative humidity, and temperature).

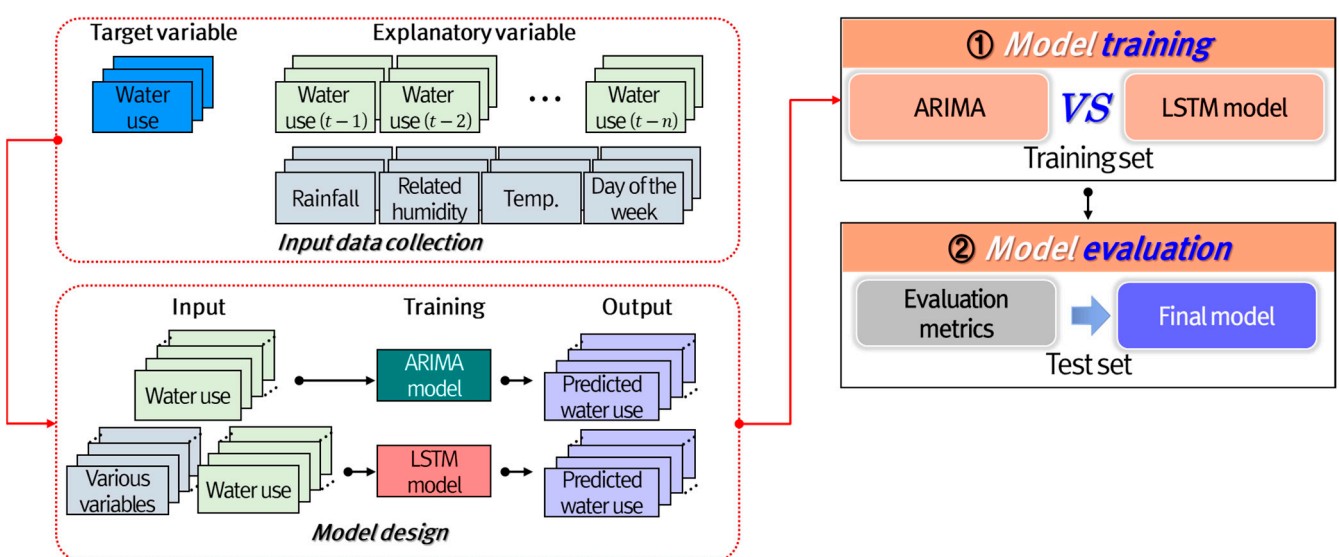

**Figure 4.** Flow chart for this study.

### 2.3.1. ARIMA Model

The AR model predicts output variables via linear dependence on the stochastic term of its previous values. This model assumes time series data as stationary, and it is expressed as Equation (1), where $p$ is the lag time of the AR model, $\varnothing$ is the AR coefficient, $y_{t-i}$ is the previous values before $t - i$ hours and $\varepsilon_t$ is the error term [39–41].

$$y_t = \varnothing_1 y_{t-1} + \varnothing_2 y_{t-2} + \cdots + \varnothing_p y_{t-p} + \varepsilon_t = \sum_{i=1}^{p} \varnothing_i y_{t-i} + \varepsilon_t \tag{1}$$

The moving average (MA) model predicts output variables using the error term ($\varepsilon_t$) of its previous values. This model assumes time series data as stationary, similar to the AR model, and it is expressed as Equation (2). Here, $q$ is the lag time of the MA model, $\mu$ is the mean of the series, and $\theta$ is the MA coefficient [42–44].

$$y_t = \mu + \varepsilon_t + \theta_1\varepsilon_{t-1} + \theta_2\varepsilon_{t-2} + \cdots + \theta_q\varepsilon_{t-q} = \mu + \sum_{i=1}^{q}\theta_i\varepsilon_{t-i} + \varepsilon_t \qquad (2)$$

The autoregressive moving-average (ARMA) model, which combines the AR and MA models to predict a more accurate output variable, is expressed as Equation (3). The model is usually referred to as the ARMA ($p$, $q$) model, where $p$ is the lag time of the AR part and $q$ is the lag time of the MA part [45–47].

$$y_t = \varnothing_1 y_{t-1} + \varnothing_2 y_{t-2} + \cdots + \varnothing_p y_{t-p} + \varepsilon_t + \theta_1\varepsilon_{t-1} + \theta_2\varepsilon_{t-2} + \cdots + \theta_q\varepsilon_{t-q} \qquad (3)$$

Box et al. [48] proposed an ARIMA model that can be used with nonstationary time series data. The nonstationary time series is converted into a stationary time series using the differencing. Here, the differencing is to make the average change in the time series constant through the difference in continuous observations. The model is usually referred to as the ARIMA ($p$, $d$, $q$) model and is expressed as Equation (4), where B is the backward shift operator, $\Delta$ is the differences, and $d$ is the parameter of the differences.

$$\varnothing_p(\text{B})\Delta^d y_t = \delta + \theta_q(\text{B})\,\varepsilon_t \qquad (4)$$

### 2.3.2. LSTM Model

Various machine learning models are applied to predict non-linear data. The LSTM model is an effective advanced neural network model when the input data is sequential data. Thus, the LSTM model was used to predict water consumption in this study.

Hochreiter and Schmidhuber invented the LSTM, a type of artificial recurrent neural network (RNN) architecture used in the field of deep learning [49]. The LSTM model was developed to address the vanishing gradient problem when training traditional RNNs, and it is widely used in studies related to time series prediction [50–52].

An LSTM unit is composed of a cell state ($C_t$), an input gate ($i_t$), an output gate ($o_t$), and a forget gate ($f_t$), as shown in Figure 5. Each state exchanges information with one another. The cell state transfers past information ($C_{t-1}$) and updated information ($C_t$) to the next cell through a forget gate, an input gate, and an output gate to update the information. The forget gate receives the new input data ($x_t$) and previous hidden state data ($h_{t-1}$) and determines what information to convey to the cell state. The input gate determines what information to update among the new information and generates a new information value ($\widetilde{C}_t$) through an activation function (tanh). The output gate determines the information to be the output. Equations (5)–(10) can be used to depict this process. Here, $\sigma$ is the activation function; $W_f$, $W_i$, $W_C$, and $W_o$ are the weights of each gate; and $b_f$, $b_i$, $b_C$, are $b_o$ are the biases of each gate.

$$f_t = \sigma\left(W_f \cdot [h_{t-1}, x_t] + b_f\right), \qquad (5)$$

$$i_t = \sigma(W_i \cdot [h_{t-1}, x_t] + b_i), \qquad (6)$$

$$\widetilde{C}_t = \tanh(W_C \cdot [h_{t-1}, x_t] + b_C, \qquad (7)$$

$$C_t = f_t \cdot C_{t-1} + i_t \cdot \widetilde{C}_t, \qquad (8)$$

$$o_t = \sigma(W_o \cdot [h_{t-1}, x_t] + b_o), \qquad (9)$$

$$h_t = o_t \cdot \tanh(C_t). \qquad (10)$$

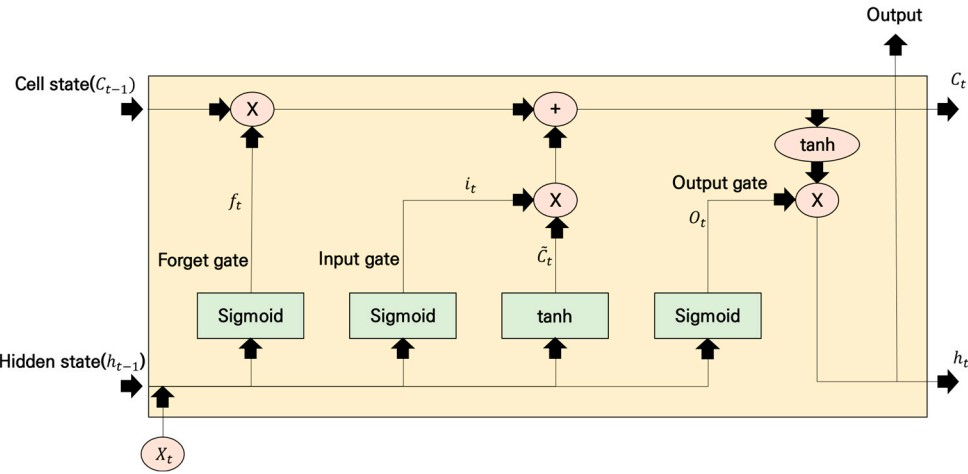

**Figure 5.** Conceptual diagram of the LSTM.

### 2.4. Evaluation Metrics

The RMSE and CC were used to evaluate the performance of each model in this study. The RMSE is used to indicate representative errors in predicted and observed values. The lower the RMSE, the better the performance, and it can be expressed as Equation (11), where n is the number of data, $y_i$ is the observed value, and $\hat{y}_i$ is the predicted value.

$$\text{RMSE} = \sqrt{\frac{1}{n} \sum_{i=i}^{n} (y_i - \hat{y}_i)^2}. \tag{11}$$

The CC measures the linear correlation between two datasets (predicted value and observed value). It is a ratio between the covariance of two datasets, and it is essentially a normalized measurement of the covariance, with the result always ranging between −1 and 1. A CC closer to −1 indicates a negative correlation, and a CC closer to +1 indicates a positive correlation. It is expressed as Equation (12), where $\overline{y}$ is the mean of the observed value and $\overline{\hat{y}}$ is the mean of the predicted value.

$$\text{CC} = \frac{\sum (y_i - \overline{y})(\hat{y}_i - \overline{\hat{y}})}{\sqrt{\sum (y_i - \overline{y})^2 \sum (\hat{y}_i - \overline{\hat{y}})^2}}. \tag{12}$$

## 3. Results

### 3.1. Application of the ARIMA Model

An autocorrelation function (ACF) analysis was conducted before training the ARIMA model (Figure 6). The autocorrelation decreased as the lag time increased, but it increased again when the lag time was seven or eight days. This result indicates that the dataset had a seven-day or eight-day cycle.

The ARIMA model consists of three parameters: an AR model parameter ($p$), an MA model parameter ($q$), and a differential parameter ($d$). In this study, the sensitivity of the parameters was analyzed to build the best ARIMA model for each type. The $p$ and $q$ parameters were considered from zero to eight with reference to the ACF results, and the d parameter was considered from zero to one. A total of 162 ($8(p) \times 8(q) \times 2(d)$) ARIMA models were developed according to parameter combination, and the ARIMA model with the lowest RMSE was selected. Table 2 summarizes the parameters of the optimal ARIMA model for each type. For Types A and D, the $p$ and $q$ parameters were set at seven, while the $d$ parameter $d$ was set at zero. For Types B and C, the $p$ and $q$ parameters were set at eight, while the $d$ parameter $d$ was set at zero.

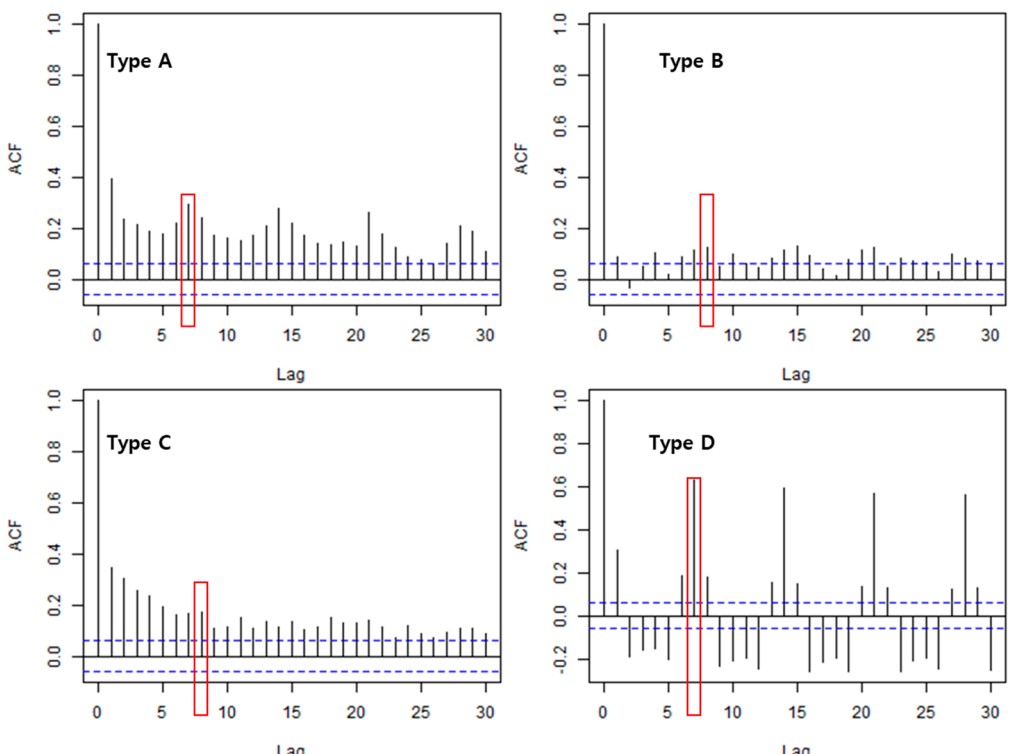

**Figure 6.** ACF results for each type.

**Table 2.** Parameters of the optimal ARIMA model for each type.

| Type | Parameter $p$ | Parameter $d$ | Parameter $q$ |
|---|---|---|---|
| A | 7 | 0 | 7 |
| B | 8 | 0 | 8 |
| C | 8 | 0 | 8 |
| D | 7 | 0 | 7 |

Table 3 summarizes the performance of the optimal ARIMA model for each type. The CC and RMSE were used to evaluate the performance of the model in the training dataset. The average correlation between the four water-use types was calculated as 93%, and the average RMSE was calculated as 4.43 m³.

**Table 3.** Performance of the optimal ARIMA model (training dataset).

| Type | Correlation | RMSE |
|---|---|---|
| A | 93.91% | 0.13 |
| B | 87.31% | 14.02 |
| C | 95.18% | 0.73 |
| D | 96.02% | 2.87 |

Figure 7 shows the prediction results of the ARIMA model in the training dataset as a time series. The solid blue lines represent the time series of the observed water consumption. At the same time, the red dashed lines represent the time series of the predicted water consumption by the ARIMA model. The solid blue lines and the red dashed lines exhibit almost similar results. Type D had the best performance in terms of performance by type, while Type B had the worst performance. This is due to Type D (school) having a relatively constant pattern, while Type B (apartment) has a complex pattern (see Figure 7).

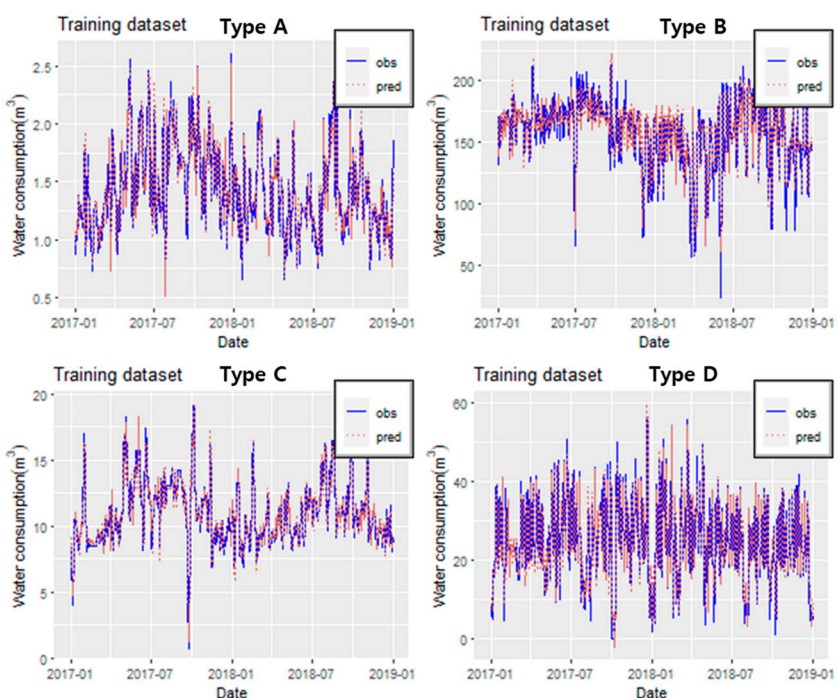

**Figure 7.** Time series of the predicted and observed values by the ARIMA model (training dataset).

Overall, the performance of the ARIMA model in the training dataset was excellent, indicating that it can make similar predictions for the observed water consumption.

### 3.2. Application of the LSTM Model

In contrast to the ARIMA model, the LSTM model can consider various explanatory variables to predict the target variable (water consumption). Water consumption data from the previous 1–7 days, weather conditions (air temperature, rainfall, and relative humidity), weekend information, and weekday information were used as explanatory variables by considering Refs. [27–29]. Table 4 shows the description of the target and explanatory variables used to train the LSTM model.

**Table 4.** Description of the target and explanatory variables.

| Variable | Abbreviation | Description |
|---|---|---|
| Target variable | $W_t$ | Water consumption corresponding to $t$ day |
| Explanatory variable | $W_{t-1}$ | Water consumption before $t-1$ day |
| | $W_{t-2}$ | Water consumption before $t-2$ day |
| | $W_{t-3}$ | Water consumption before $t-3$ day |
| | $W_{t-4}$ | Water consumption before $t-4$ day |
| | $W_{t-5}$ | Water consumption before $t-5$ day |
| | $W_{t-6}$ | Water consumption before $t-6$ day |
| | $W_{t-7}$ | Water consumption before $t-7$ day |
| | T | Daily air temperature |
| | R | Daily rainfall |
| | Rh | Daily relative humidity |
| | W.d | Weekday and weekend |

A sensitivity analysis was also performed on the LSTM model to derive the optimal parameters. The parameters of the LSTM model mainly deal with "units," which refers to the number of chains, "batch size", which refers to the number of data extracted for

learning, and "epoch", which refers to repetitive learning [12]. Here, the epoch and batch size are determined to repeat until they maximize learning performance according to the number of data, and units are determined to accommodate the information to the maximum according to the number of explanatory variables. In this study, the parameters were considered 6, 12, 24, and 36 for the units; 12, 36, 72, and 144 for the batch size; and 20, 30, 50, and 100 for the epoch; and grid search was performed for this list. In addition, activation function was considered as "Tanh", and "dropout layer" was considered as 45% to minimize overfitting problem. On the other hand, the other parameters such as learning rate, momentum were set to their default values. The LSTM models were developed using 64 parameter combinations for each type, and the LSTM model with the best performance was selected. Table 5 shows the parameter combinations for the top four models. The batch size and epoch in the top four models are 12 and 100, respectively.

**Table 5.** Parameter combinations for the top four models.

| Model | Units | Batch Size | Epoch |
|:---:|:---:|:---:|:---:|
| Model 1 | 6 | 12 | 100 |
| Model 2 | 12 | 12 | 100 |
| Model 3 | 24 | 12 | 100 |
| Model 4 | 36 | 12 | 100 |

Figure 8 shows the loss graph for each type using the top four models. Here, the red line and points represent model 1, the blue line and points represent model 2, the yellow line and points represent model 3, and the green line and points represent model 4. The x-axis represents the epoch, and the y-axis represents the loss per epoch. When the LSTM models were trained, the RMSE was used as the loss metric, and the target variable was normalized. Thus, the loss unit in Figure 8 differs from the original consumption data unit. In addition, we used 20% of the batch size as a validation to prevent overfitting, and the loss presented in Figure 8 is the validation loss.

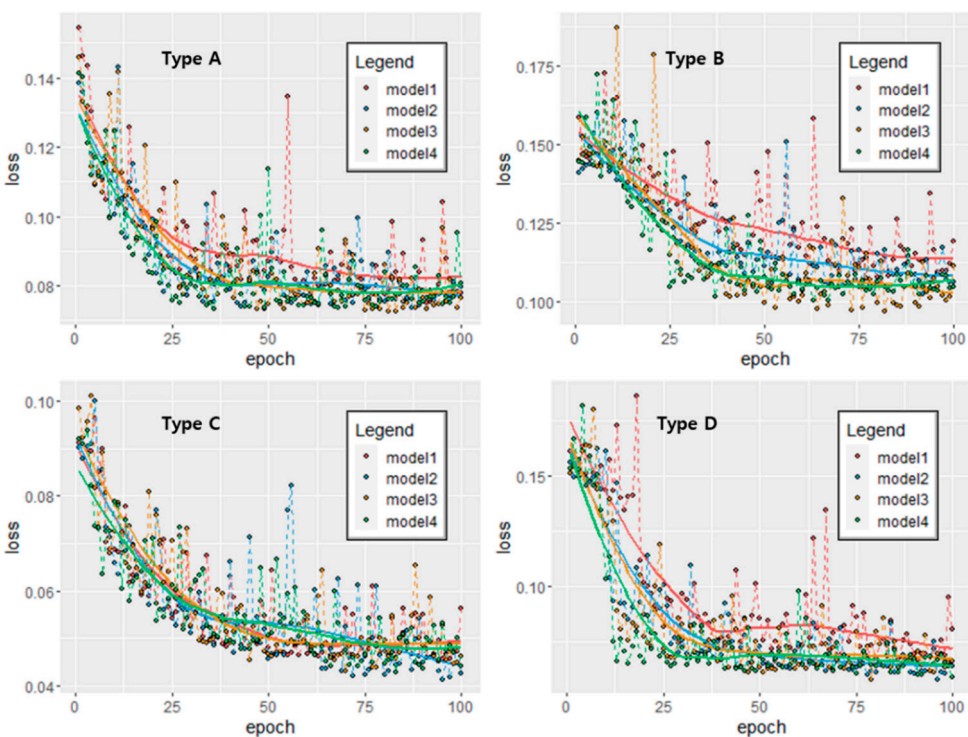

**Figure 8.** Loss graph for each type using the top four models.

Figure 8 shows that as the epoch increased, the loss graph for all types decreased and converged, and the loss trend did not increase again, indicating that the overfitting problem did not occur. Thus, the loss was the smallest when Type A was model 3, Type B was model 3, Type C was model 2, and Type D was model 4. The optimal LSTM model for each type was determined through this result. Table 6 shows the performance of the optimal LSTM model in the training dataset, and Figure 9 shows the time series of the LSTM model's prediction results. The average correlation for the four types was 89%, and the average RMSE was calculated to be 5.6 m$^3$. It can also be seen that Type D has the highest correlation and Type B has the lowest correlation. All the LSTM models had a correlation of 80% or higher in the training dataset. Thus, the LSTM models were well-trained well for observing water consumption data.

**Table 6.** Performance of the optimal LSTM model (training dataset).

| Type | Correlation | RMSE |
|------|-------------|------|
| A | 90.29% | 0.17 |
| B | 80.96% | 17.07 |
| C | 92.00% | 0.93 |
| D | 93.71% | 4.24 |

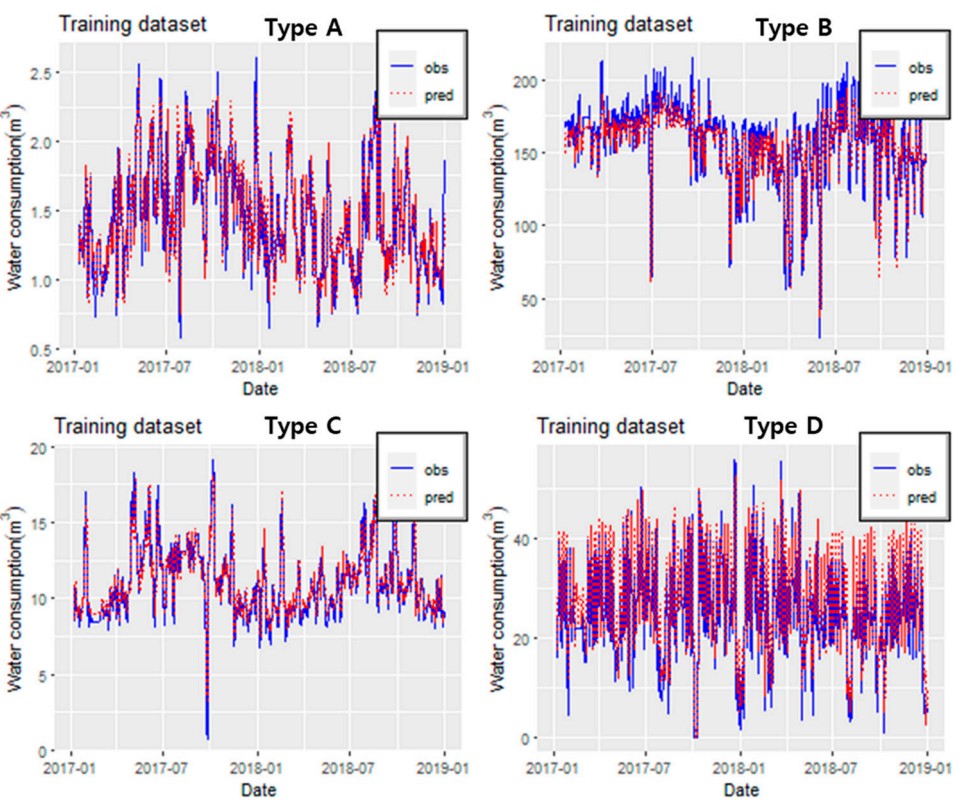

**Figure 9.** Time series of the predicted and observed values of the LSTM model (training dataset).

When the performance of the ARIMA model (see Table 3) was compared to that of the LSTM model, it was discovered that the performance of the LSTM model was slightly lower. Therefore, the two models had to be evaluated in a test dataset that was not used for model learning.

### 3.3. Performance Evaluation of Each Model

The two models were evaluated in the test dataset (1 January 2019 to 31 December 2019). Figure 10 shows the time series of the predicted result by each model. The solid

blue line represents the observed water consumption time series, the red dashed line represents the predicted time series of the LSTM model, and the solid green line represents the predicted time series of the ARIMA model. The red dashed line tends to be similar to the solid blue line, but the solid green line tends to be more underestimated than the solid blue line. In addition, the red dashed line appears to be more similar to the solid blue line than the green line. Table 7 summarizes the performance evaluation of each model in the test dataset. The ARIMA model had an average correlation of 62% and an average RMSE of 8.91 m$^3$. The correlation decreased by about 31%, and the RMSE increased by 4.48 m$^3$ in comparison to the results in the training dataset. Compared to that in the training dataset, the performance of the ARIMA model in the test dataset decreased due to overfitting. However, the LSTM model had an average correlation of 89% and an average RMSE of 5.60 m$^3$. These results were similar to the performance in the training dataset. The LSTM model had a better correlation than the ARIMA model by an average of 27%.

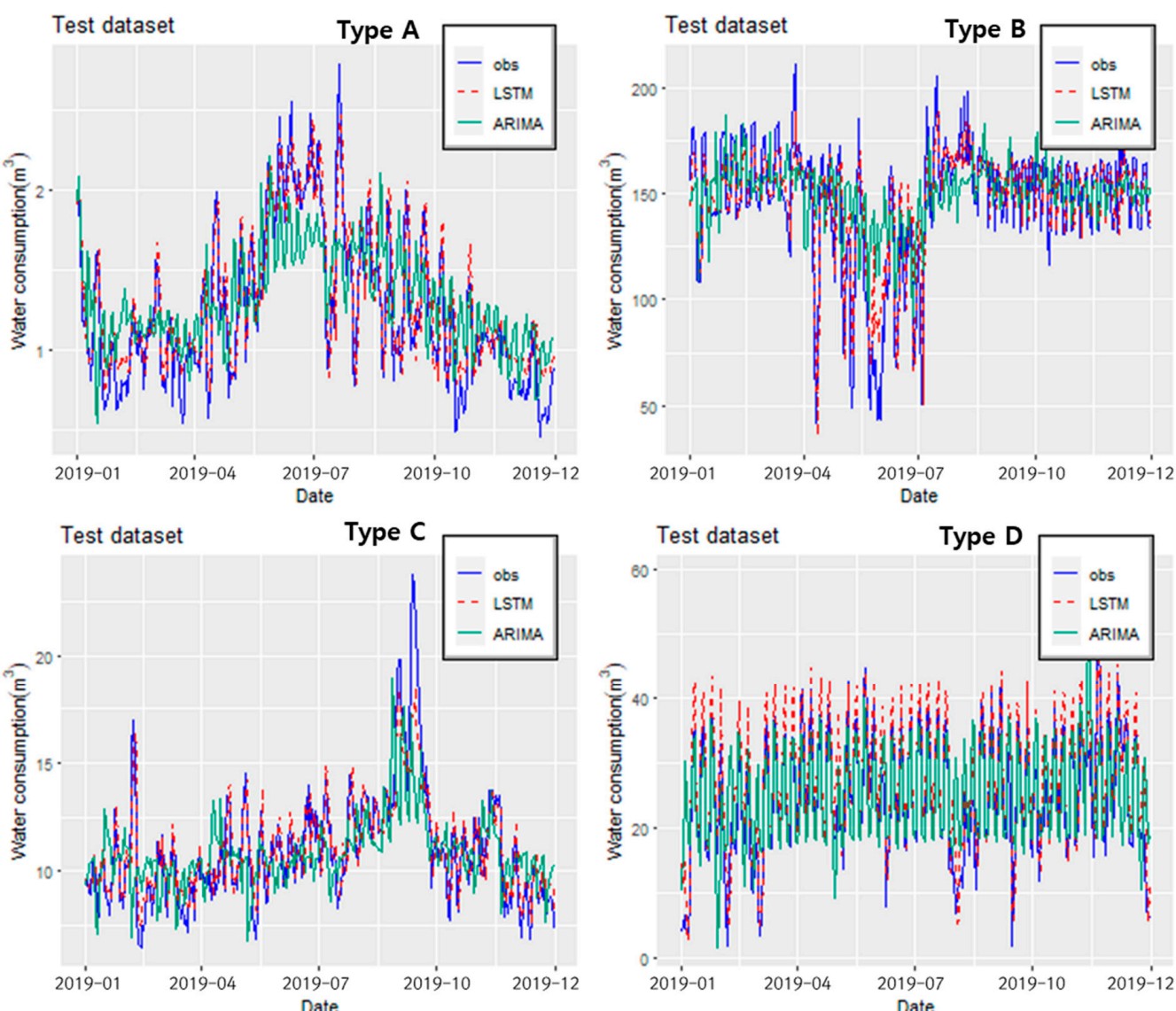

**Figure 10.** Time series of the predicted and observed values in the test dataset for each model.

**Table 7.** Performance evaluation of each model (test dataset).

| | ARIMA | | LSTM | |
|---|---|---|---|---|
| **Type** | **Correlation** | **RMSE** | **Correlation** | **RMSE** |
| A | 65.81% | 0.36 | 92.70% | 0.19 |
| B | 55.42% | 26.37 | 82.96% | 17.58 |
| C | 56.42% | 2.21 | 89.15% | 1.24 |
| D | 69.79% | 6.71 | 91.29% | 4.75 |

Taylor diagrams are mathematical diagrams designed to graphically indicate the RMSE, CC and Standard deviation (SD) [53]. Figure 11 shows a Taylor diagram for each region. Green dotted line indicates RMSE, blue dotted line indicates SD, black dotted line indicates CC, and the green square point is SD of observation. In general, the smaller the RMSE and SD, and the closer CC is to one, the better the model performance. Here, the points that are closer to the green square point in this diagram mean better model performance, and the red and blue points represent the performance of ARIMA and LSTM in the diagram. As a result, blue points are closer to the green square point than red points. Therefore, the LSTM model outperforms the ARIMA model in predicting water consumption at the household level as it can be trained with various patterns.

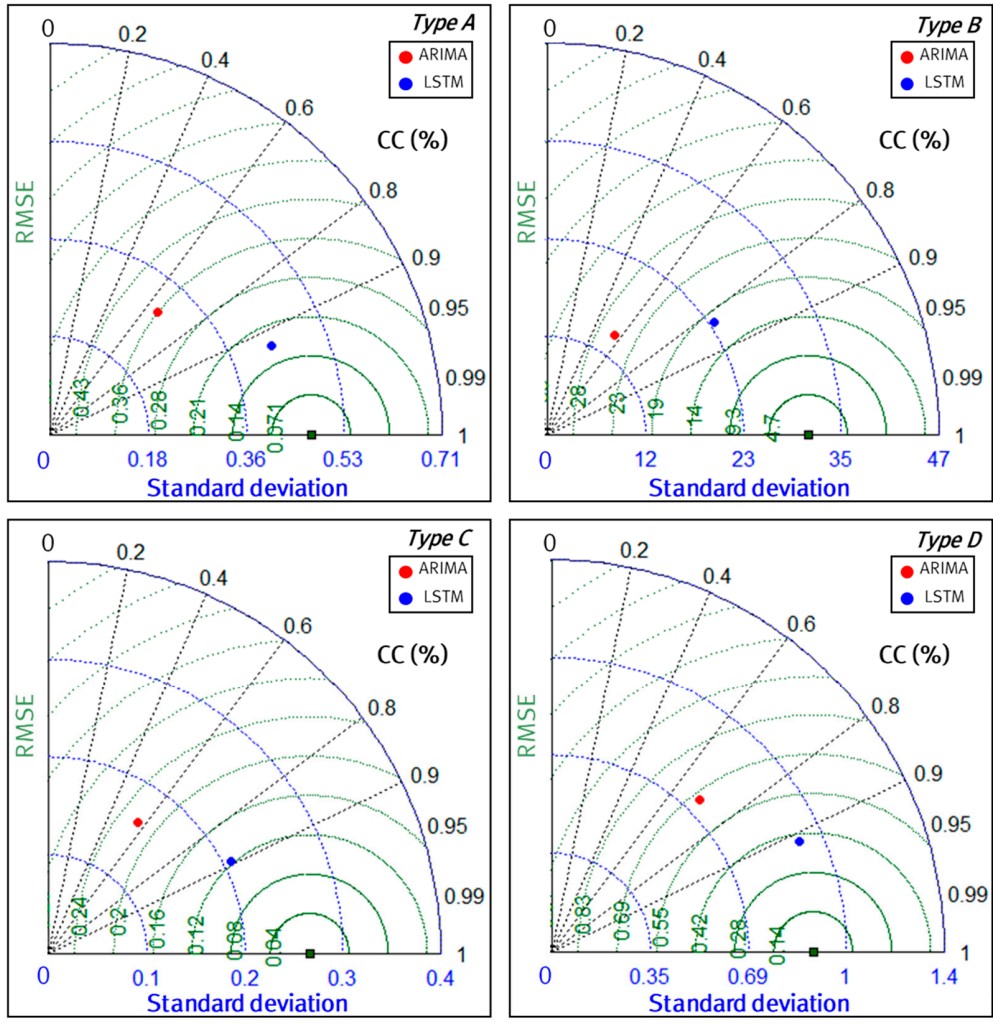

**Figure 11.** Taylor diagram for each type.

## 4. Conclusions and Discussion

This study developed a model to predict water consumption for four different water-use types (A: detached house, B: apartment, C: restaurant, and D: elementary school) using smart water meter data. The ARIMA model, which is a traditional time series prediction model, and the deep learning-based LSTM model were used as the prediction models. The LSTM model considered both previous water consumption data and external factors such as weather and information on weekends and weekdays as explanatory variables. Types A, B, and C demonstrated a pattern of using much water on weekends, whereas Type D demonstrated a pattern of using much water on weekdays.

The analysis revealed that the ARIMA model outperformed the LSTM model by a slight difference in the training dataset, but both models demonstrated excellent performance. However, in the test dataset, the ARIMA model's correlation decreased by about 31% on average, while the LSTM model demonstrated excellent performance (CC: 89% and RMSE: 5.60 $m^3$), similar to the results in the training dataset. These results indicate that the ARIMA model has an overfitting problem and does not accurately learn the non-linear characteristics of each water-use type. However, the proposed model was trained very well for each water-use type by employing a deep learning model to learn nonlinear characteristics and consider external factors. In addition, the proposed model can maintain performance even when new data is added. Therefore, this data can inform customers about the predicted water consumption fee or manage water efficiently.

Since the proposed model was designed to predict water consumption one day later, it cannot predict water consumption after one week or one month. This model also considers external factors such as weather and weekend information. Still, it has limitations in that it does not consider national holidays in Korea, such as Chuseok, and human activities. Therefore, the model will be improved in future research by designing it to consider various human activities and national holidays and predict water consumption over long periods, such as a week or a month. The target area will then be expanded using this model for all consumers at the household level in a single city. Therefore, it is believed that future studies will further improve the usability of the model.

**Author Contributions:** Conceptualization, J.K. and H.L.; Data curation, H.L. and D.K.; Formal analysis, J.K.; Methodology, J.K., M.L. and D.K.; Supervision, M.L., H.H. and H.S.K.; Writing—original draft, J.K. and H.L.; Writing—review and editing, H.H. and H.S.K. All authors have read and agreed to the published version of the manuscript.

**Funding:** INHA UNIVERSITY Research Grant.

**Institutional Review Board Statement:** Not applicable.

**Informed Consent Statement:** Not applicable.

**Data Availability Statement:** Not applicable.

**Acknowledgments:** This work was supported by INHA UNIVERSITY Research Grant.

**Conflicts of Interest:** The authors declare no conflict of interest.

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
