# Peer review of "Development of a Deep Learning-Based Prediction Model for Water Consumption at the Household Level"

_water, doi:10.3390/w14091512_

Round 1
Reviewer 1 Report
The paper deals with the important issue of the development of a deep learning-based prediction model for water consumption at the household level. The study used a deep learning-based long short-term memory (LSTM) approach to develop a water consumption prediction model for each customer. The proposed model considers several variables to learn nonlinear water consumption patterns. An ARIMA model and an LSTM model in the training dataset for customers with four different water-use types (detached houses, apartment, restaurant, and elementary school) were developed. Remarks: In section 2 the model is mentioned, but there is no detail about how these analyses have been carried out. In order for the contribution to be useful and serve as a reference, the methodology should be thoroughly presented to sustain results.
Author Response
Thank you for your favorable review of this manuscript.

Reviewer 2 Report
This paper reports the application of a deep learning-based prediction model (long short-term memory approach) to predict household-level water consumption. The model results are compared with those from a classical ARIMA model. This paper is interesting and publishable. This referee recommends a moderate revision before accepting this manuscript for publication.
It is fine to have the classical AR model selected as a baseline for the model comparison, which is limited in prediction for the non-linear systems. A number of studies reported comparison studies of non-linear approaches. Please comment on why the LSTM is adopted over other approaches. Authors are suggested to give a brief summary of recent development in the use of machine-learning for non-linear applications (e.g. Abba, S. I., Gaya, M. S., Yakubu, M. L., Zango, M. U., Abdulkadir, R. A., Saleh, M. A., ... & Wahab, N. A. (2019, June). Modelling of uncertain system: a comparison study of linear and non-linear approaches. In 2019 IEEE International Conference on Automatic Control and Intelligent Systems (I2CACIS) (pp. 1-6). IEEE.).
Please explain how the values of hyperparameters of the model are selected (epoch, batch size, unit). Some have been described but some others could be equally important. E.g. a number of nodes and hidden layers, weight initialization, activation function selection, learning rate, momentum, dropout, etc. Did any grid search or randomized search algorithms apply? How the choices of the hyperparameters relate to the model accuracy? Any optimization procedure was applied to minimize the over-fitting problem?
Please add a reference for the choice of explanatory variables. Please comment on the contribution of each parameter to the prediction result accuracy.
Table 4. Please describe the explanatory variables T, R, and Rh. Daily mean? On which day?
Author Response
Thank you very much for your insightful and detail review. We have revised attentively the manuscript in order to include your comments. We believe that this manuscript is substantially improved through the result of the revision. Please see our point-by-point response (in blue) to your comments (in red).

Reviewer 3 Report
This paper deals with the application of a deep learning scheme for water consumption forecast at household level. This method has been extensively applied in the water distribution network framework for predicting water consumption where a large number of data are available but it has been never applied in death in the South Korea context.
The study is properly design: the methodology proposed (LSTM-based model) is compared with a standard approach (ARIMA model) and all comparisons are properly design and the conclusions are accurately described. I think, consequently, that this work is worth publishing in the WATER journal.
I have just a minor comments:
Could the author specify for sake of clarity what y_t stands (Eq 1,2,3)?
Author Response
Thank you very much for your favorable review. Please see our point-by-point response (in blue) to your comments (in red).

Round 2
Reviewer 2 Report
I am pleased to recommend publication, with a minor amendment of grammar check. To name an example, in lines 343-344, "On other hands, the other parameters such as were set to their default 343 values.". It may be read as: "On the other hand, ..." Also, what are the other parameters referred to, for the "such as"?
Author Response
Thank you for your detail comment, i revised line 343-344 to reflect your comment.
L343-344 (Clean Ver.): On the other hand, the other parameters such as learning rate, momentum were set to their default values.